# Biosynthesis of thiocarboxylic acid-containing natural products

Liao-Bin Dong [1], Jeffrey D. Rudolf [1], Dingding Kang[2], Nan Wang[1], Cyndi Qixin He[3], Youchao Deng[2], Yong Huang[2], K. N. Houk[3], Yanwen Duan[2] & Ben Shen [1,4,5]

Thiocarboxylic acid-containing natural products are rare and their biosynthesis and biological significance remain unknown. Thioplatensimycin (thioPTM) and thioplatencin (thioPTN), thiocarboxylic acid congeners of the antibacterial natural products platensimycin (PTM) and platencin (PTN), were recently discovered. Here we report the biosynthetic origin of the thiocarboxylic acid moiety in thioPTM and thioPTN. We identify a thioacid cassette encoding two proteins, PtmA3 and PtmU4, responsible for carboxylate activation by coenzyme A and sulfur transfer, respectively. ThioPTM and thioPTN bind tightly to β-ketoacyl-ACP synthase II (FabF) and retain strong antibacterial activities. Density functional theory calculations of binding and solvation free energies suggest thioPTM and thioPTN bind to FabF more favorably than PTM and PTN. Additionally, thioacid cassettes are prevalent in the genomes of bacteria, implicating that thiocarboxylic acid-containing natural products are under-appreciated. These results suggest that thiocarboxylic acid, as an alternative pharmacophore, and thiocarboxylic acid-containing natural products may be considered for future drug discovery.

[1] Department of Chemistry, The Scripps Research Institute, Jupiter, FL 33458, USA. [2] Xiangya International Academy of Translational Medicine, Central South University, Changsha, Hunan 410013, China. [3] Department of Chemistry and Biochemistry, University of California, Los Angeles, CA 90095, USA. [4] Department of Molecular Medicine, The Scripps Research Institute, Jupiter, FL 33458, USA. [5] Natural Products Library Initiative at The Scripps Research Institute, The Scripps Research Institute, Jupiter, FL 33458, USA. Correspondence and requests for materials should be addressed to B.S. (email: shenb@scripps.edu)

Thiocarboxylic acids are an underappreciated pharmacophore in drug discovery and development. In contrast to carboxylic acids, which are one of the most important and biologically active pharmacophores in modern therapeutics[1], the significance of thiocarboxylic acids is often overlooked due to their intrinsic instability, difficult preparation, and rare occurrence in nature[2–4]. In addition, the scarcity of thiocarboxylic acid-containing natural products, of which there are only two, thioquinolobactin (TQB)[5,6] and pyridine-2,6-dithiocarboxylic

acid (PDTC)[7], has hindered their biosynthetic and biological studies (Supplementary Fig. 1a).

Thioplatensimycin (thioPTM, **1**) and thioplatencin (thioPTN, **2**) are newly discovered thiocarboxylic acid-containing congeners of the antibacterial natural products platensimycin (PTM, **3**) and platencin (PTN, **4**) (Fig. 1a–c)[8]. PTM and PTN were originally isolated from *Streptomyces platensis* MA7327 and MA7339, respectively, representing a class of promising natural product antibiotics that target bacterial and mammalian fatty acid

**Fig. 1** In vivo characterization of thiocarboxylic acid biosynthesis. **a** Genetic organization of the *ptm* gene clusters from the dual PTM–PTN producers *S. platensis* MA7327, *S. platensis* CB00739, and *S. platensis* CB00765. **b** Genetic organization of the *ptn* gene cluster from the PTN-exclusive producer *S. platensis* MA7339. The thioacid cassette investigated in this study, *ptmU4* and *ptmA3*, are present and highlighted (red rectangle) in both the *ptm* and *ptn* gene clusters. **c** Structures of thioplatensimycin (**1**, thioPTM), thioplatencin (**2**, thioPTN), platensimycin (**3**, PTM), and platencin (**4**, PTN). The aliphatic ketolide and 3-amino-2,4-dihydroxybenzoic acid (**5**, ADHBA) moieties are highlighted in blue and red, respectively. **d** Structures of ADHBA (**5**) and 3-amino-2,4-dihydroxythiobenzoic acid (ADHBSH, **5-SH**). **e** UV at 280 nm from LC-MS analysis of metabolites from SB12039 (Δ*ptmA3*), SB12040 (Δ*ptmU4*), SB12041 (Δ*ptmS1*), SB12042 (Δ*ptmS2*), and SB12043 (Δ*ptmS4*) using the PTM-PTN dual overproducer, SB12029, as a positive control. **f** Extracted ion chromatograms (EIC, *m/z* at both 170 and 186) from LC-MS analysis of metabolites from heterologous reconstitution of **5-SH** in model *Streptomyces* hosts. SB12306 was individually scanned using *m/z* 170 and 186. std standard

synthases[9–11]. Structurally, they are comprised of two distinct moieties, a diterpene-derived lipophilic ketolide and a 3-amino-2,4-dihydroxybenzoic acid (ADHBA, **5**), linked by an amide bond (Fig. 1c, d)[12,13]. We recently developed a dual PTM–PTN over-producing strain, SB12029, through the inactivation of the negative transcriptional regulator *ptmR1* in the *ptm* gene cluster of *Streptomyces platensis* CB00739 (Fig. 1a, b), which is now used as a model strain for the study of PTM and PTN biosynthesis[14,15]. When following previously reported procedures for the production of **3** and **4** in SB12029, we discovered that **1** and **2** were also produced in high titers[8]. Although **1** and **2** were not isolated, we unambiguously established that **1** and **2** possessed thiocarboxylic acid moieties by a combination of high-resolution electrospray ionisation mass spectrometry and chemical transformation[8]. A sulfur-containing PTM pseudodimer, PTM D1, was also isolated supporting the presence of **1** and **2** in SB12029[8]. The discovery of **1** and **2**, and natural congeners thereof (Supplementary Fig. 1b)[16,17], questioned whether **3** and **4** were final biosynthetic products of the *ptm* and *ptn* biosynthetic gene clusters and set the stage to study thiocarboxylic acid biosynthesis and explore the chemistry and biology of thiocarboxylic acid-containing natural products.

Here we report the (i) production of **1** and **2** by both wild-type strains and engineered overproducers known to produce both **3** and **4** or only **4**, supporting the legitimacy of the thioacid congeners as natural products; (ii) identification and in vivo and in vitro characterization of a thioacid cassette, which, in combination with the sulfur-carrier protein machinery, is responsible for thiocarboxylic acid biosynthesis; (iii) discovery that the thioacid cassette is broadly distributed in nature; (iv) biological implications of **1** and **2** as antibiotics; and (v) preliminary substrate promiscuity studies of the thioacid cassette, highlighting its potential as a pair of biocatalysts for thiocarboxylic acid synthesis. This work provides clear biochemical evidence supporting that the thioacid cassette, which encodes PtmA3 and PtmU4, is responsible for carboxylate activation by coenzyme A (CoA) and sulfur transfer, respectively, and works together with the sulfur-carrier protein trafficking system for thioPTM and thioPTN biosynthesis. Furthermore, given the subtleties in properties between carboxylic and thiocarboxylic acids and the proven activities of thioPTM and thioPTN, thiocarboxylic acid and thiocarboxylic acid-containing natural products in general should now be considered as an alternative pharmacophore in future drug discovery efforts.

## Results

***ptm* and *ptn* gene clusters producing thioPTM and thioPTN**. As **1** and **2** were initially discovered from SB12029, we were curious if the thiocarboxylic acid congeners were simply a result of the overproducing nature of the strain. We selected eight strains, four wild-type strains and four engineered overproducers known to produce both **3** and **4** or only **4**. The wild-type strains *S. platensis* MA7327[9] and MA7339[10], *S. platensis* CB00739, and *S. platensis* CB00765 (an alternative PTM–PTN dual producer)[15], and the overproducing strains SB12001 (MA7327 Δ*ptmR1*)[18], SB12600 (MA7339 Δ*ptmR1*)[16], SB12026 (CB00739 Δ*ptmR1*)[15], and SB12027 (CB00765 Δ*ptmR1*)[15] were fermented under conditions known for **1** and **2** production (Supplementary Methods). Liquid chromatography-mass spectrometry (LC-MS) analysis of all eight strains revealed the production of both **1** and **2** (along with **3** and **4**) in the PTM–PTN dual producers or **2** (along with **4**) in PTN-exclusive producers (Supplementary Fig. 2a, b). A time-course of SB12029 fermentation revealed that only **1** and **2** were detected on day 1, with increasing amounts of **3** and **4** as the fermentation continued (Supplementary Fig. 2c, d). These findings support that **1** and **2**, rather than **3** and **4**, might be the bona

fide natural products of the PTM and PTN biosynthetic machineries.

Comparison of the *ptm* and *ptn* gene clusters (Fig. 1a, b)[15,19] with gene clusters known for biosynthesis of the thiocarboxylic acid- or thioester-containing natural products TQB (*qbs*)[5,6], PDTC (*pdt*)[7], and yatakemycin (YTK, *ytk*)[20] revealed two conserved genes encoding an acyl-CoA synthetase [PtmA3, ~30% protein sequence identity with QbsL (32%), OrfJ (28%, from *pdt*), and YtkG (32%)] and a type III CoA-transferase [PtmU4, ~36% identity with QbsK (37%), OrfI (36%, from *pdt*), and YtkF (35%)]. We individually inactivated *ptmA3* and *ptmU4* in the PTM–PTN dual overproducer SB12029 (Supplementary Tables 1 and 2, Supplementary Data 1, and Supplementary Figs. 3–6). The resultant mutants SB12039 (Δ*ptmA3*) and SB12040 (Δ*ptmU4*) lost their ability to produce **1** and **2** but still produced **3** and **4** (Fig. 1e). The production of **3** and **4** in SB12039 and SB12040 implies that (i) **3** and **4** are directly converted into **1** and **2**, respectively, or (ii) **5** is converted into **5-SH** prior to its coupling with the ketolides (Fig. 2a, b)[19,21]. To verify the timing of thiocarboxylic acid formation in the biosynthesis of **3** and **4**, we introduced *ptmB1*, *ptmB2*, and *ptmB3*, which encodes the biosynthesis of **5** (Fig. 2a)[21], together with *ptmA3* and *ptmU4* into four model *Streptomyces* hosts to afford *S. albus* SB12303, *S. lividans* SB12304, *S. coelicolor* SB12305, and *S. avermitilis* SB12306 (Supplementary Fig. 7). LC-MS analysis revealed that, while SB12303–SB12305 did not produce either **5** or **5-SH**, SB12306 produced both **5** and **5-SH** (Fig. 1f). Production of **5-SH** in the heterologous host *S. avermitilis* confirms that PtmA3 and PtmU4 are required for thiocarboxylic acid biosynthesis, while the sulfur donor for thiocarboxylic acid formation is conserved between *S. platensis* and *S. avermitilis* (Supplementary Table 3), and likely among all *Streptomyces* species.

**Utilizing sulfur-carrier proteins as the sulfur donor**. Thiocarboxylic acids at the C-termini of small sulfur-carrier proteins are utilized as direct sulfur donors for the biosynthesis of essential cofactors including thiamine and molybdenum cofactor in nearly all bacteria species[22,23]. The biosynthesis of this protein thiocarboxylate species is well-established and occurs via nucleophilic attack of a protein-tethered persulfide on an adenylated carboxylate of a C-terminal glycine (G)[22,23]. Given that **5-SH** was readily produced in *S. avermitilis*, and the fact that the *tqb* and *pdt* gene clusters possess their own cognate members of the sulfur-carrier protein system[5–7], we suspected that the native sulfur donor for **1** and **2** is from similar primary metabolism machinery. Since no homologs are present in the *ptm* gene cluster, we searched the genome of CB00739 for potential candidates. Four sulfur-carrier proteins (PtmS2, PtmS3, *Sp*SCP1, and *Sp*SCP2), one MoaZ homolog (PtmS4), and one JAMM family metalloprotease (PtmS1) were encoded within the genome (Supplementary Fig. 8). Both PtmS2 and PtmS3 have capped C-termini with either two residues (MV) or a single residue (C) following the GG motif, respectively (Supplementary Fig. 9); *Sp*SCP1 and *Sp*SCP2 have exposed C-terminal GG sequences. Typically, sulfur-carrier proteins with capped C-termini require a protease, as exemplified by QbsD in TQB biosynthesis and whose biochemical function was previously reported, to expose the C-terminal GG sequence for subsequent activation and sulfur transfer[24]. Since the protease-encoding *ptmS1* was found clustered with the capped sulfur-carrier protein-encoding *ptmS2* in CB00739 (Supplementary Fig. 8), we hypothesized that PtmS2 may be utilized for the biosynthesis of **1** and **2**. Similar two-gene cassettes were found in the *tqb* and *pdt* gene clusters[5–7], as well as in the genome of *S. avermitilis* (Supplementary Table 3).

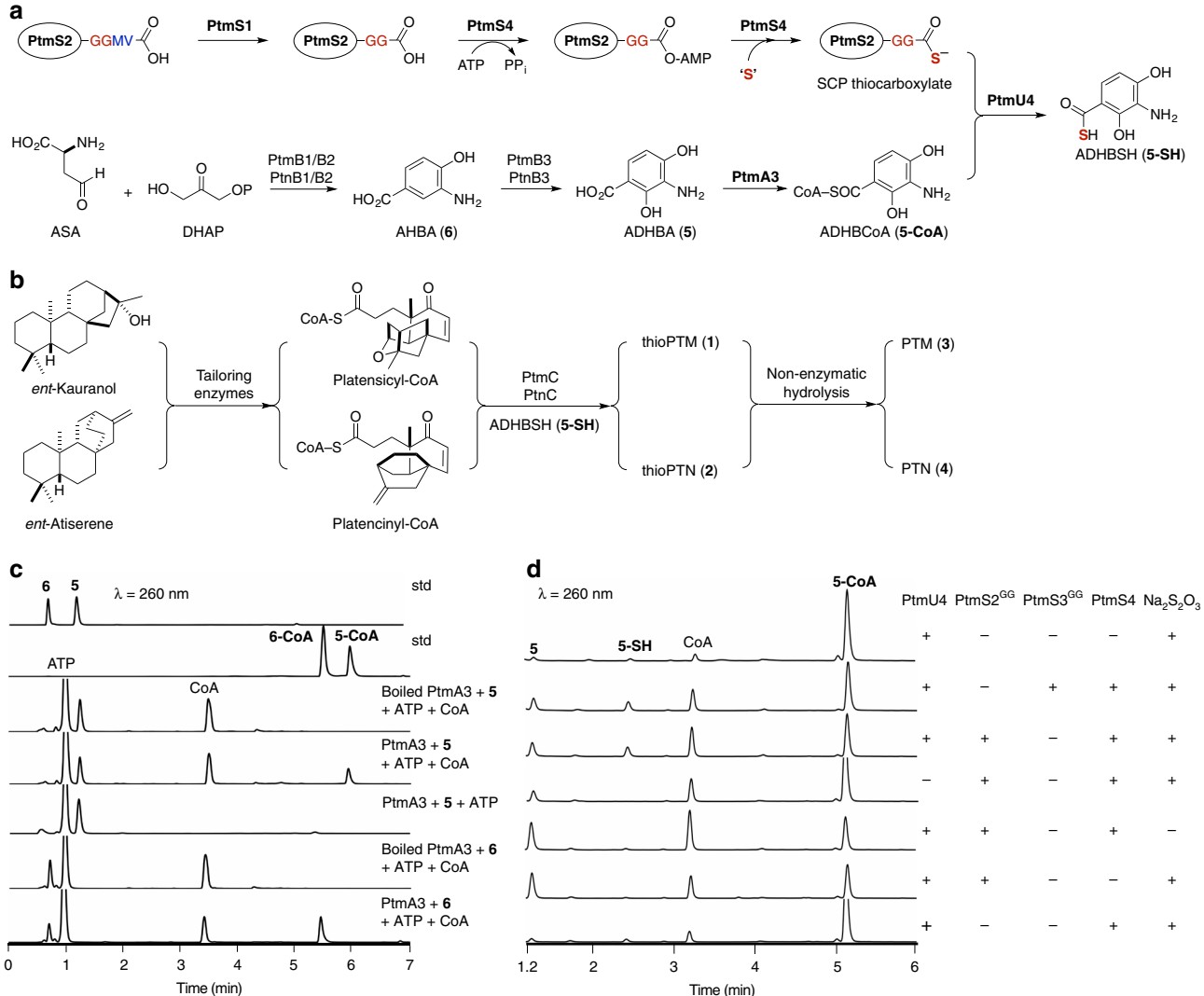

**Fig. 2** Proposed biosynthetic pathway and in vitro characterization of thiocarboxylic acid biosynthesis. **a 5-SH** is the enzymatic product of the transformation of **5** by the thioacid cassette, PtmA3 and PtmU4, capturing a sulfur atom from the sulfur-carrier protein machinery. The C-terminal GG motif and capped residues (MV in PtmS2) are highlighted in red and blue, respectively. SCP sulfur-carrier protein (circled), ASA aspartate semialdehyde, DHAP dihydroxyacetone phosphate, AHBA (**6**) 3-amino-4-hydroxybenzoic acid, ADHBCoA (**5-CoA**) $S$-(3-amino-2,4-dihydroxybenzoate) coenzyme A. **b** ThioPTM (**1**) and thioPTN (**2**) result from the coupling of **5-SH** with platensicyl- and platencinyl-CoA, respectively. **c** UV at 260 nm from HPLC analysis of in vitro PtmA3 reactions with **5** and **6**. std standard. **d** UV at 260 nm from HPLC analysis of in vitro PtmU4 reactions with **5-CoA** using the native sulfur donor. Sodium thiosulfate (Na$_2$S$_2$O$_3$) was used as the inorganic sulfur source

We individually inactivated *ptmS1*, *ptmS2*, and *ptmS4* in SB12029, yielding mutants SB12041 (Δ*ptmS1*), SB12042 (Δ*ptmS2*), and SB12043 (Δ*ptmS4*), respectively (Supplementary Figs. 10–15). Similar to SB12039 and SB12040, both SB12041 and SB12043 lost their ability to produce **1** and **2** but still produced **3** and **4** (Fig. 1e). In contrast, SB12042 retained its ability to produce **1** and **2** (Fig. 1e), indicating that PtmS2 may be functionally redundant (e.g., complemented by PtmS3)[25]. The inability of SB12041 and SB12043 to produce **1** and **2** corroborates the use of capped sulfur-carrier proteins (i.e., PtmS2 and PtmS3), as opposed to proteins with pre-exposed GG motifs (i.e., *Sp*SCP1 and *Sp*SCP2), by the PTM and PTN biosynthetic machineries.

**Reconstituting thiocarboxylic acid biosynthesis in vitro**. Upon incubation of **5**, ATP, and CoA in the presence of PtmA3 (Supplementary Fig. 16a, c), high-performance liquid chromatography (HPLC) analysis of the reaction mixture revealed the disappearance of **5** and concomitant production of $S$-(3-amino-2,4-dihydroxybenzoate) coenzyme A (ADHBCoA, **5-CoA**, Fig. 2a, c and Supplementary Figs. 17 and 18). Substitution of **5** with **3** or **4** under the same assay conditions failed to afford any new products, confirming that **3** and **4** are not the precursors of **1** and **2**, respectively (Supplementary Fig. 19). Steady-state kinetics of PtmA3, using **5** or 3-amino-4-hydroxybenzoic acid (AHBA, **6**, Fig. 2a, c, Supplementary Figs. 20a, b, 21, and 22), revealed 12-fold higher catalytic efficiency ($k_{cat} K_m^{-1}$) for **5** over **6** (Table 1), supporting **5** as the native substrate of PtmA3. When **5-CoA**, ATP, and either PtmS2$^{GG}$ or PtmS3$^{GG}$ [the C-terminal caps were removed during cloning[24] as the native sulfur donor, supplemented with sodium thiosulfate (Na$_2$S$_2$O$_3$)[23,25], were incubated with PtmS4 and PtmU4 (Supplementary Fig. 16a–c), LC-MS analysis of the assay mixture revealed the appearance of free CoA and **5-SH** (Fig. 2d). Replacement of the native sulfur donor with potassium hydrosulfide (KSH) as a sulfur donor surrogate[26] led to increased **5-SH** formation (Supplementary Fig. 23a), isolation of

**Table 1 Summary of the steady-state kinetics of PtmA3**

| Substrate | $K_m$ (μM) | $K_i$ (μM) | $K_{cat}$ (min$^{-1}$) | rel $K_{cat} K_m^{-1}$ |
|---|---|---|---|---|
| **5** | 22.9 ± 3.0 | 184 ± 24 | 14.1 ± 0.9 | 1 |
| **6** | 470 ± 60 | 638 ± 89 | 23.8 ± 2.2 | 0.082 |
| **7** | $(1.24 ± 0.11)×10^3$ | n.a. | 6.89 ± 0.3 | 0.009 |
| **8** | 70.9 ± 5.0 | n.a. | 1.03 ± 0.02 | 0.024 |
| **9** | 290 ± 21 | n.a. | 1.39 ± 0.03 | 0.008 |
| **10** | 322 ± 15 | n.a. | 15.8 ± 0.3 | 0.079 |

All experiments were performed in triplicate and the data are listed with standard deviations The relative rates are compared to the native substrate **5**. *n.a.* not applicable

which allowed unambiguous structural characterization by $^1$H and $^{13}$C NMR (Supplementary Figs. 24 and 25).

**PtmU4 catalyzing sulfur transfer.** PtmU4 was bioinformatically predicted to possess two type III CoA-transferase domains. CoA-transferases, which typically perform reversible CoA transfer reactions from CoA-thioester donors to various organic acid CoA acceptors are classified into three subgroups (types I–III) based on differences in their protein sequences and reaction mechanisms (Supplementary Fig. 26)[27]. Canonical type III CoA-transferases are proposed to utilize a conserved Asp residue to form mixed anhydride and covalent thioester intermediates with the CoA donor and CoA moiety, respectively; the liberated donor free acid is not released prior to binding the CoA acceptor (Fig. 3a and Supplementary Figs. 26a and 27)[27,28]. Type III CoA-transferases can be further separated into one-domain and two-domain proteins. Members of the two-domain family have tandem CoA-transferase domains, which have highest homology at the N-termini of the individual domains, separated by a poorly conserved linker[29]. DddD, the only characterized two-domain type III CoA-transferase, dually functions as a CoA-transferase and a lyase mediating the first two steps of dimethyl sulfide release from dimethylsulfoniopropionate (Supplementary Fig. 26c)[30]. Phylogenetic analysis of selected one- and two-domain type III CoA-transferases from bacteria revealed that PtmU4/PtnU4 clades with other two-domain type III CoA-transferases and separated from all one-domain type III CoA-transferases and DddD (Fig. 3b)[29]. DddD forms an outgroup from both the one-domain and two-domain type III CoA-transferases.

Sequence alignment of PtmU4 and its homologs revealed that they indeed possess a conserved Asp at the C-terminal CoA transferase domain (Fig. 3a and Supplementary Fig. 27). To determine if PtmU4 requires this conserved Asp for thiocarboxylic acid formation, Asp430 was mutated to Ala, Glu, or Asn by site-directed mutagenesis (Supplementary Fig. 16d). Circular dichroism indicated that the structures of these mutants were not significantly perturbed relative to native PtmU4 (Supplementary Fig. 16e and Supplementary Methods). The activities of each mutant were dramatically reduced (125–400-fold), supporting Asp430 as a key residue in PtmU4 catalysis (Supplementary Fig. 23b). Although trace amounts of **5-SH** were identified in the negative control reaction without a native sulfur donor (Fig. 2d), the ~3-fold rate enhancement of sulfur transfer by PtmU4 using either PtmS2$^{GG}$ or PtmS3$^{GG}$ (Fig. 2d), along with the near loss of activity in the Asp430 mutants, confirms genuine enzyme catalysis.

**Thiocarboxylic acids as a potential pharmacophore.** If **1** and **2** are the bona fide final metabolites of the PTM and PTN biosynthetic machineries, it is fascinating that **3** and **4**, non-enzymatic hydrolyzed metabolites of the PTM and PTN pathway

(Fig. 2a, b), possess such exquisite biological activity. We isolated and spectroscopically characterized **1** and **2** (Supplementary Tables 4 and 5 and Supplementary Figs. 28–37), and tested their antibacterial activities against *Staphylococcus aureus* ATCC 25923 and *Kocuria rhizophila* (previously *Micrococcus luteus*) ATCC 9431 (Supplementary Methods)[31]. Both **1** and **2** retained strong antibacterial activities with minimum inhibitory concentrations (MICs) ranging from 1 to 4 μg mL$^{-1}$ (2–8-fold higher than those of **3** and **4**, Table 2 and Supplementary Fig. 38a). As the ADHBA moiety of **3** and **4** makes key interactions with the target proteins FabF and FabH[9,10], the ADHBSH moieties of **1** and **2** likely bind in a similar manner. Binding assays, using the *Escherichia coli* FabF C163Q mutant, which mimics the acyl-enzyme intermediate[9], revealed that the dissociation constants ($K_D$) for **1** and **2** were ~2-fold tighter than those of **3** and **4** (Table 2 and Supplementary Figs. 38b and 39). Density functional theory (DFT) calculations were performed on the ADHBA moiety of **3** and **4** and the ADHBSH moiety of **1** and **2** with two interacting residues, H303 and H340, in the *E. coli* FabF C163Q mutant (PDB 2GFX, Supplementary Data 3)[9]. The carboxylate of **5** forms stronger hydrogen bonds ($\Delta G = -12.0$ kcal mol$^{-1}$) with the imidazoles of the His residues than the thiocarboxylate of **5-SH** (two conformations, each $\Delta G \approx -9.6$ kcal mol$^{-1}$, Fig. 4a). In a three-residue model (H303, H340, Q163), the geometry of the **5-SH-b** conformation is favored over **5-SH-a** by 3.5 kcal mol$^{-1}$; consistent with the two His model, ADHBA binding is favored over ADHBSH (**5-SH-b**) by 2.5 kcal mol$^{-1}$ (Fig. 4b). Calculation of solvation free energies of **5** (–58.2 kcal mol$^{-1}$) and **5-SH** (–55.1 kcal mol$^{-1}$) revealed that **5-SH** is less well-solvated in water (Fig. 4c). Therefore, although **5** forms a more stable complex, **5-SH** exhibits a higher activity (favored binding) due to a net energy difference[32] of 0.6 kcal mol$^{-1}$.

**Biocatalysts for thiocarboxylic acid synthesis.** We first examined the promiscuity of PtmA3 and PtmU4 using a series of aryl acids with KSH as the sulfur donor. Both PtmA3 and PtmU4 showed broad substrate promiscuity by efficiently catalyzing the CoA activation and thiolation, respectively, to afford the corresponding aryl thiocarboxylic acids (Tables 1 and 3, Fig. 5a, and Supplementary Figs. 20c–f and 40–48). When catalysis is coupled, PtmA3 and PtmU4 convert carboxylic acids into thiocarboxylic acids in the presence of a catalytic amount of CoA (Fig. 5b, c). This biocatalytic platform provides a practical solution for thiocarboxylic acid synthesis of small molecules with future opportunities to expand the substrate scope through enzyme engineering.

**Discussion**

Since the discovery of PTM and PTN as promising antibiotics over a decade ago, these natural products have garnered considerable attention. Significant progress in understanding the biosynthetic pathways of **3** and **4** has been made through the use of microbial genomics[11]. Given the amount of time and resources spent on studying **3** and **4**, we were initially surprised to identify sulfur-containing congeners of PTM and PTN[8,17]. The discovery of the thiocarboxylic acid-containing **1** and **2** raised a provocative question: which natural products are the true biosynthetic end products of the *ptm* and *ptn* biosynthetic gene clusters? The current study clearly demonstrates that (i) **1** and **2** are produced by several wild-type and engineered strains of *S. platensis*; the four wild-type strains were isolated from three different continents (Africa, Europe, and Asia)[9,10,15], (ii) **1** and **2** is produced prior to the detection of **3** and **4**, and in certain conditions, **3** and **4** are non-enzymatic hydrolysis products of **1** and **2**, (iii) the biosynthesis of the thiocarboxylic acid moieties in **1** and **2** is

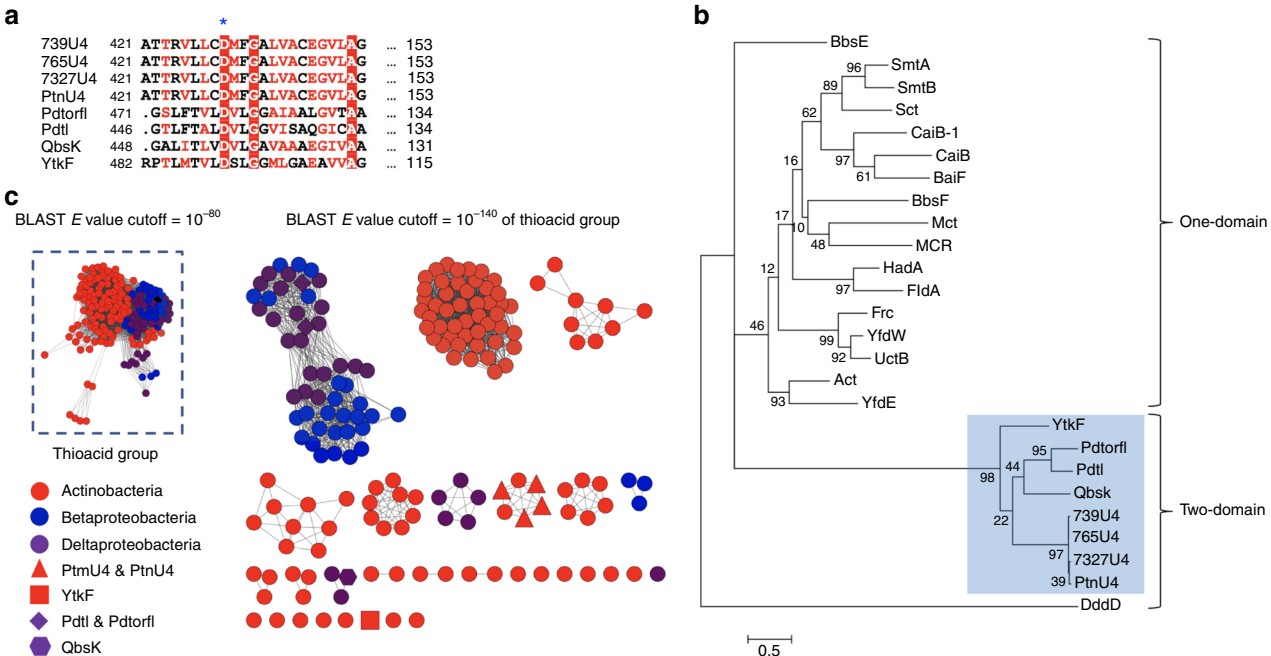

**Fig. 3** Bioinformatics analysis of type III CoA-transferases from bacteria. **a** Sequence alignment of selected PtmU4 homologs from bacteria. Aligned residues are colored based on the level of conservation (red box with white character and red character show strict identity and similarity, respectively). The conserved catalytic residue, aspartic acid (D430), is shown with blue asterisks. A full sequence alignment was included in Supplementary Fig. 27. **b** Phylogenetic analysis of the selected type III CoA-transferase from bacteria. The sequences used in the phylogenetic tree include selected one-domain and two-domain type III CoA-transferases (Supplementary Methods). **c** A sequence similarity network (SSN) of thioacid group. The BLAST e-value threshold is $10^{-140}$ with median 58% sequence identity over 500 residues. All PtmU4 homologs in the thioacid group are from three classes of bacteria: Actinobacteria, Betaproteobacteria, and Gammaproteobacteria. Each node represents protein sequences sharing 100% sequence identity. Colors represent different classes in bacteria. Shapes in **c** represent the thiocarboxylic acid biosynthesis related type III CoA-transferases discussed in the paper. A complete SSN including all 2401 two-domain type III CoA-transferases in bacteria is shown in Supplementary Fig. 49

**Table 2 Antibacterial activities and FabF binding experiments with compounds 1–4**

| | MIC (µg mL$^{-1}$) | | Binding assay with *E. coli* FabF C163Q | | |
|---|---|---|---|---|---|
| | S. aureus | K. rhizophila | $k_a$ (M$^{-1}$ s$^{-1}$) | $k_d$ (s$^{-1}$) | $K_D$ (M) |
| thioPTM (**1**) | 4 | 1 | $(3.0 \pm 0.1) \times 10^5$ | $(1.3 \pm 0.1) \times 10^{-3}$ | $(4.4 \pm 0.3) \times 10^{-9}$ |
| thioPTN (**2**) | 1 | 0.5 | $(2.3 \pm 1.2) \times 10^5$ | $(6.9 \pm 1.7) \times 10^{-3}$ | $(3.3 \pm 0.9) \times 10^{-8}$ |
| PTM (**3**) | 0.5 | 1 | $(5.8 \pm 0.9) \times 10^5$ | $(5.1 \pm 0.9) \times 10^{-3}$ | $(8.8 \pm 1.4) \times 10^{-9}$ |
| PTN (**4**) | 0.25 | 0.25 | $(6.2 \pm 2.2) \times 10^5$ | $(4.5 \pm 2.6) \times 10^{-2}$ | $(7.0 \pm 1.7) \times 10^{-8}$ |
| Minimum inhibitory concentrations (MICs) were determined in triplicate using the broth dilution method. Binding assays are reported as means with standard deviations of at least two replicates | | | | | |

genetically encoded by at least four genes, two of which are located inside the *ptm* gene cluster, and (iv) PtmA3 and PtmU4 convert **5** into **5-SH** in vitro and within the context of the native sulfur donors. While our data supports that **1** and **2**, rather than **3** and **4**, are the bona fide biosynthetic end products of the PTM and PTN biosynthetic machineries, we could not rule out if **3** and **4** are the legitimate natural products that would be produced in its native environment (i.e., the complex environment of soil).

The identification and characterization of the thioacid cassette and its functional integration in the biosynthesis of **1** and **2**, along with its presence in the biosynthetic gene clusters of the thiocarboxylic acid- or ester-containing natural products PDTC, TQB, and YTK, led us to examine the prevalence of the thioacid cassette in bacterial genomes. A database search (as of September 5, 2017) of putative thioacid cassettes in bacteria produced 2,401 PtmU4 homologs (i.e., two-domain type III CoA-transferases), most of which are from Actinobacteria and Proteobacteria (Supplementary Fig. 49). In a sequence similarity network[33] of

PtmU4 homologs, a thioacid group, including PtmU4 from various PTM and PTN producers and the homologs involved in the biosynthesis of PDTC, TQB, and YTK, was found at an e-value threshold of $10^{-80}$ (Fig. 3c). In contrast to the relatively conserved PtmU4 homologs from Proteobacteria, the homologs from Actinobacteria have more sequence diversity, forming 28 different clusters (Fig. 3c). Among the 175 sequences in the thioacid group, 160 (>90%) were encoded in genetic proximity (≤2 genes) to homologs of PtmA3 (Supplementary Data 2 and Supplementary Fig. 50), confirming the broad distribution of the thioacid cassette in nature. Although it is possible that these cassettes are nonfunctional genetic remnants of evolution, the observable functional integration of the thioacid cassettes in thioPTM, thioPTN, PDTC, TQB, and YTK biosynthesis lead us to speculate that thiocarboxylic acid-containing natural products, or their derivatives thereof, are vastly underrepresented among known natural products.

The isolation of **1** and **2** gave us an opportunity to assess the biological implications of thiocarboxylic acid-containing natural

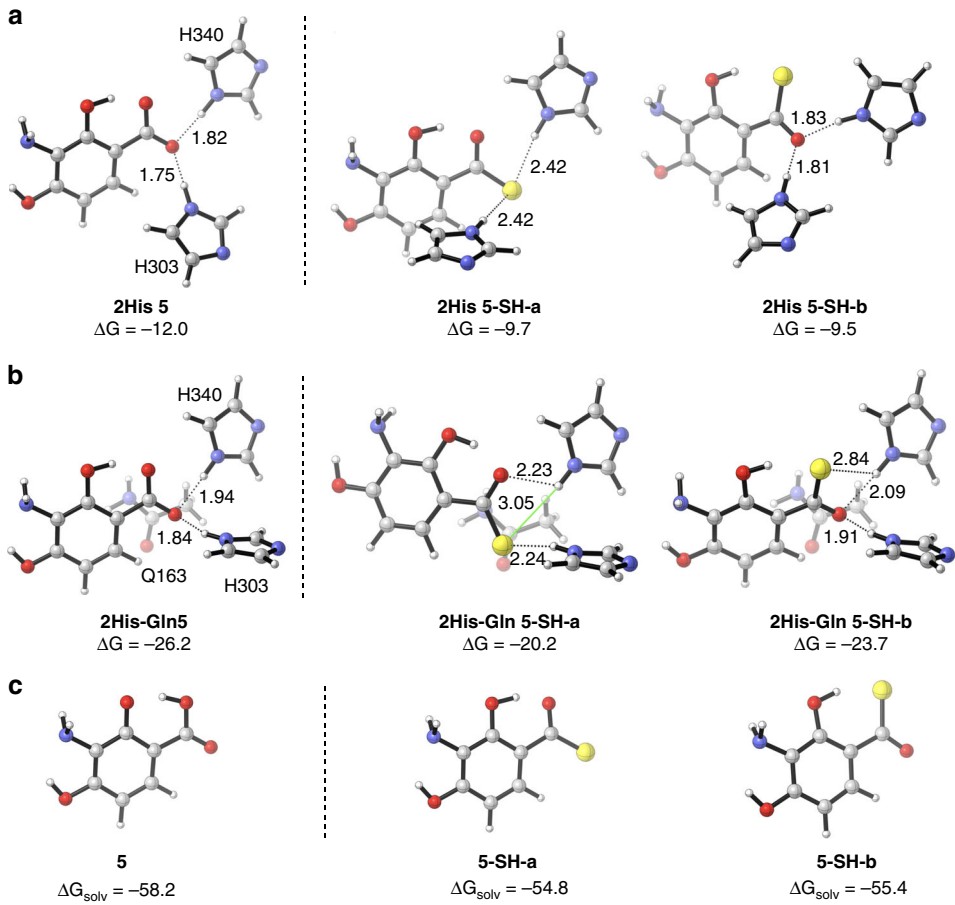

**Fig. 4** Density functional theory calculations of binding and solvation free energies of **5** and **5-SH**. **a** Optimized geometries and energies of **5** and **5-SH** (modeled as truncated versions of **1–4**) in the presence of two imidazoles (the side chains of H303 and H340) in the gas phase. The units of bond length (shown as dotted lines) and Gibbs energies are Å and kcal mol$^{-1}$, respectively. **b** Optimized geometries and energies of **5** and **5-SH** bound to the sides chains of three fixed amino acids (H303, H340, Q163) in the gas phase. **c** Free energies of solvation of **5** and **5-SH**

| Table 3 Summary of the relative activities of PtmU4 | | |
|---|---|---|
| **Substrate** | **Rate (s$^{-1}$)** | **Relative rate (%)** |
| **5-CoA** | $(2.43 \pm 0.07) \times 10^{-3}$ | $100 \pm 3$ |
| **6-CoA** | $(2.46 \pm 0.02) \times 10^{-3}$ | $101 \pm 1$ |
| **7-CoA** | $(5.82 \pm 0.08) \times 10^{-3}$ | $240 \pm 3$ |
| **8-CoA** | $(1.50 \pm 0.2) \times 10^{-4}$ | $6 \pm 1$ |
| **9-CoA** | $(5.23 \pm 0.2) \times 10^{-4}$ | $22 \pm 1$ |
| **10-CoA** | $(2.44 \pm 0.1) \times 10^{-3}$ | $100 \pm 6$ |

All experiments were performed in triplicate and the data are listed with standard deviations. The relative rates are compared to the native substrate **5-CoA**. *S*-(3-amino-4-chlorobenzoate) coenzyme A (**7-CoA**); *S*-(2-amino-3-hydroxy benzoate) coenzyme A (**8-CoA**); *S*-(2-amino-4-fluorobenzoate) coenzyme A (**9-CoA**); *S*-(6-hydroxy-2-naphthalenecarboxylate) coenzyme A (**10-CoA**)

products. PTM and PTN inhibit the decarboxylating condensing enzymes FabF and FabH in bacterial type II fatty acid synthesis (FASII)[9,10]. The carboxylic acid moiety of **3** and **4** mimics the malonyl-acyl carrier protein substrate and ionically interacts with two His residues in the Cys-His-His catalytic triad, resulting in competitive inhibition[9–11]. Structure–activity relationship studies of **3** and **4** have concluded that modification of their ADHBA moieties results in loss of antibiotic activity[11,21,34]. The thiocarboxylic acid congeners **1** and **2**, whose structures only differ from **3** and **4** at one atom, retained strong antibacterial activities and were found to bind slightly tighter to FabF. Our DFT calculations were congruent with **1** and **2** being better binders of

FabF than **3** and **4** while exhibiting minimal differences in their observed MICs. Similarly, TQB and PDTC, which are bacterial siderophores, have improved activities compared to their carboxylic acid congeners[6,7]. It is still unclear if and why *S. platensis* produces **1** and **2** in nature, but the biosynthetic role of the thioacid cassette and its prevalence in bacterial genomes would suggest that thiocarboxylic acid-containing natural products might have important and unsolved biological roles in nature. In conclusion, thiocarboxylic acids, which have been an underappreciated pharmacophore in drug discovery and development, should now be considered in future studies.

## Methods

**Bacterial strains, plasmids, and chemicals**. Strains, plasmids, and PCR primers used in this study are listed in Supplementary Information. PCR primers were obtained from Sigma-Aldrich. Q5 high-fidelity DNA polymerase, restriction endonucleases, and T4 DNA ligase were purchased from NEB and used by following the protocols provided by the manufacturers. DNA gel extraction and plasmid preparation kits were purchased from Omega Bio-Tek. DNA sequencing was conducted by Eton Bioscience. The REDIRECT Technology kit for PCR-targeting homologous recombination was provided by The John Innes Center (Norwich, UK)[35]. pOJ260 was used as a shuttle vector for gene homologous recombination[36]. *E. coli* ET12567/pUZ8002 was used as the host for intergeneric conjugations[37]. pUWL201PWT, which is a derivative of pUWL201PW[38] containing an *oriT* sequence that was cloned into its *Pst*I site, was used as the shuttle vector for gene complementations, biotransformation, and heterologous production of PtmU4 in *Streptomyces*. Cosmid libraries were screened by PCR using OneTaq 2× Master Mix with GC buffer (NEB). For Southern analysis, digoxigenin labeling of DNA probes, hybridization, and detection were performed according to the protocols provided by the manufacturer (Roche Diagnostics Corp.). *S. platensis* CB00739[15], CB00765[15], MA7327[9], and MA7339[10], and their pathway-specific

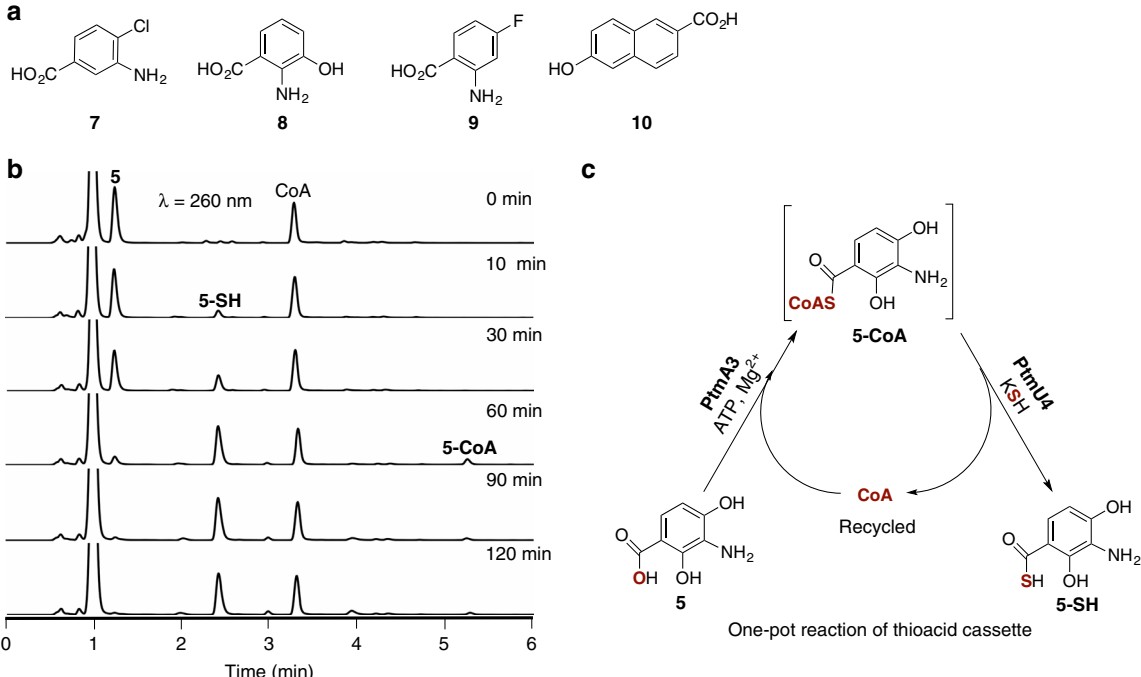

**Fig. 5** Biocatalysis of thiocarboxylic acids using PtmA3 and PtmU4. **a** Chemical structures of the aryl acids tested. **b** UV at 260 nm of time-course analysis of the one-pot reaction to synthesize **5-SH** from **5** using a combination of PtmA3 and PtmU4 using recycled coenzyme A (CoA). In this reaction, 1 equivalent of **5** was supplied with 0.5 equivalent of CoA. After 120 min, **5** was completely consumed and transformed into **5-SH**, indicating CoA was recycled during the reaction. **c** Chemical scheme for this one-pot reaction transforming carboxylic acid to thiocarboxylic acid. The functional groups involved in this transformation are highlighted in red. 3-Amino-4-chlorobenzoic acid (**7**); 2-amino-3-hydroxybenzoic acid (**8**); 2-amino-4-fluorobenzoic acid (**9**); 6-hydroxy-2-naphthalenecarboxylic acid (**10**)

negative regulator *ptmR1* inactivation mutants, SB12026[15], SB12029[14], SB12027[15], SB12001[18], and SB12600[16] were reported previously. *S. albus* J1074[39], *S. lividans* K4-114[40], *S. avermitilis* SUKA22[41], and *S. coelicolor* M1146[42] were used as model *Streptomyces* hosts for small-molecule biotransformation and protein production. Other common chemicals, biochemical, and media components were purchased from standard commercial sources.

**In-frame deletion of *ptmA3* in SB12029 to afford SB12039**. To construct the plasmid for in-frame deletion of *ptmA3*, two 3-kb fragments of the genes upstream and downstream of *ptmA3* were amplified from cosmid pBS12037, a cosmid containing a partial *ptm* gene cluster[15], with the primers 739A3up_F, 739A3up_R, 739A3down_F, and 739A3down_R. Both fragments were cloned into the *Hin*dIII and *Eco*RI sites of pOJ260 to obtain pBS12075. pBS12075 was transformed into *E. coli* ET12567/pUZ8002 and introduced into *S. platensis* SB12029 by intergeneric conjugation[35]. After several rounds of passaging the exconjugants, double crossovers via homologous recombination were selected by the apramycin-sensitive phenotype. The genotype of the in-frame deletion mutant SB12039 was verified by PCR analysis and Southern analysis.

**Inactivation of *ptmU4* in SB12029 to afford SB12040**. The *ptmU4* gene was replaced with the *aac(3)IV+oriT* resistance cassette from pIJ773 using λRED-mediated PCR-targeting mutagenesis[35] in *E. coli* BW25113/pIJ790 harboring pBS12037, a cosmid containing a partial *ptm* gene cluster[15]. The genotype of the resultant Δ*ptmU4* mutant cosmid, pBS12074, was confirmed by PCR analysis using primers 739U4ID_F and 739U4ID_R. pBS12074 was transformed into the non-methylating *E. coli* ET12567/pUZ8002 and introduced into *S. platensis* SB12029 by intergeneric conjugation. Single crossovers of Δ*ptmR1*/Δ*ptmU4* were selected by screening for apramycin resistance on ISP4 medium. After another round of passaging the single-crossover exconjugants in solid ISP4 medium, the Δ*ptmR1*/Δ*ptmU4* mutant SB12040, a result of double-crossover homologous recombination, was selected for by screening for an apramycin-resistant and kanamycin-sensitive phenotype. The genotype of SB12040 was confirmed by PCR and Southern analysis.

Inactivation of *ptmS1*, *ptmS2*, and *ptmS4* and disruption of *ptmS3* was performed using the protocol described above for the Δ*ptmR1*/Δ*ptmU4* mutant SB12040. Each genotype was verified by PCR analysis and Southern analysis.

**Heterologous production of 5-SH in model *Streptomyces* hosts**.
pUWL201PWT was used as an *E. coli*–*Streptomyces* expression shuttle vector to construct a **5-SH** production system in model *Streptomyces* hosts. The candidate thioacid cassette genes, *ptmA3* and *ptmU4*, were amplified by PCR using the primers 739U4pUW_F and 739U4pUW_R, and 739A3pUW_F and 739A3pUW_R from pBS12037 and individually cloned into pET-44b(+). *ptmU4* was cloned into the *Nde*I and *Pst*I sites and *ptmA3*, placed downstream of *ptmU4*, was cloned into the *Pst*I and *Hin*dIII sites to yield pBS12084. The constructed fragment of *ptmU4*–*ptmA3* was cut from pBS12084 at the *Nde*I and *Hin*dIII sites and cloned into pUWL201PWT at the same sites to construct pBS12085.

The three ADHBA biosynthetic genes, *ptmB1*, *ptmB2*, and *ptmB3*, were amplified as a single fragment by PCR using the primers 739B1B3pUW_F and 739B1B3pUW_R from pBS12037, which was subsequently cloned into the *Hin*dIII and *Eco*RI sites of pBS12085. The resulting construct, pBS12086, possessed *ptmU4*-*ptmA3*-*ptmB1*-*ptmB2*-*ptmB3* (Supplementary Figure 7). pBS12086 was transformed into *E. coli* ET12567/pUZ8002 and introduced into four *Streptomyces* model strains (*S. albus* J1074, *S. lividans* K4-114, *S. avermitilis* SUKA22, and *S. coelicolor* M1146) by intergeneric conjugation. Clones containing pBS12086 were selected with thiostrepton.

PTM fermentation medium, supplemented with thiostrepton, was used for the production of **5-SH** in the model *Streptomyces* hosts. After fermentation for 2 days at 28 °C, the fermentation broth was directly used for LC-MS analysis.

**Gene cloning**. The *ptmA3*, *ptmU4*, *ptmS2*^GG, *ptmS3*^GG, and *ptmS4* genes from *S. platensis* CB00739 were amplified by PCR from genomic DNA with Q5 DNA polymerase (NEB). The PCR product was purified, treated with T4 polymerase, and cloned into pBS3080[43] according to ligation-independent procedures to afford pBS12087 (harboring *ptmA3*), pBS12088 (harboring *ptmU4*), pBS12089 (harboring *ptmS4*), pBS12090 (harboring *ptmS2*^GG, amino acid residues 1–90), and pBS12091 (harboring *ptmS3*^GG, amino acid residues 1–91). The *E. coli fabF* gene containing a site-directed mutation resulting in FabF C163Q was cloned into pBS3080[43] as described above, resulting in pBS12096. pUWL201PWT was used as an *E. coli*–*Streptomyces* expression shuttle vector and protein expression of PtmU4 in *Streptomyces*. The full-length *ptmU4* gene together with an N-terminal His₆-tag sequence was amplified by PCR from pBS12088 using the 739StrU4_F and 739StrU4_R primers. Thus, *ptmU4* was cloned into the *Nde*I and *Hin*dIII sites of pUWL201PWT affording pBS12092. For site-directed mutagenesis of *ptmU4*, the *ptmU4* gene from pBS12092 was amplified in two steps by primer extension[44] using the 739StrU4_F and 739Stru4_R primers with internal primers containing the desired mutation. The mutant *ptmU4* genes were then cloned into pUWL201PWT as described above yielding pBS12093–pBS12095.

**Gene expression and protein production and purification**. PtmA3, PtmS2$^{GG}$, PtmS3$^{GG}$, and PtmS4 were produced in *E. coli*. For enzyme activity assays, the plasmid harboring each gene was transformed into *E. coli* BL21(DE3) (Life Technologies) and grown in 1 L of lysogeny broth (LB) at 37 °C with shaking at 250 rpm until an $OD_{600}$ of 0.6 was reached. The culture was cooled to 4 °C, gene expression was induced with the addition of 0.25 mM isopropyl β-D-1-thioga-lactopyranoside, and the cells were grown around 18 h at 18 °C with shaking. After harvesting the cells by centrifugation at 4000 *g* for 15 min at 4 °C, the pellet was resuspended in lysis buffer (50 mM Tris, pH 8.0, containing 300 mM NaCl and 10 mM imidazole), lysed by sonication, and centrifuged at 15,000 *g* for 30 min at 4 °C. The supernatant was purified by nickel-affinity chromatography using an ÄKTAxpress system (GE Healthcare Life Sciences) equipped with a HisTrap column. The resultant protein with an N-terminal His$_6$-tag was desalted using a HiPrep desalting column (GE Healthcare Biosciences) and concentrated using an Amicon Ultra-15 concentrator (Millipore) in 50 mM Tris, pH 7.8, containing 100 mM NaCl, 50 mM KCl, and 5% glycerol. Protein concentrations were determined from the absorbance at 280 nm using a molar absorptivity constant of each protein. Individual aliquots of each protein were stored at –80 °C until use.

PtmU4 was produced in *S. avermitilis* SUKA22 for enzyme activity assays. pBS12092 was transformed into *E. coli* ET12567/pUZ8002 and introduced into *S. avermitilis* SUKA22 by intergeneric conjugation. Positive colonies were selected using thiostrepton and named *S. avermitilis* SB12307. Fresh spores of SB12307 were inoculated into TSB seed medium supplemented with thiostrepton and cultured for 2 days. Three liters of TSB medium was inoculated with 5% (v/v) seed culture supplemented with thiostrepton and incubated at 28 °C and 250 rpm for 2 days. After harvesting the cells by centrifugation at 3750 *g* for 30 min at 4 °C, the pellet was resuspended in lysis buffer (50 mM Tris, pH 8.0, containing 300 mM NaCl and 10 mM imidazole) and 1 mg mL$^{-1}$ lysozyme and 1.5 tablets of Protease Inhibitor Cocktail (Roche) were added. After incubation on ice for 2 h, the pellet was lysed by sonication, and centrifuged at 15,000 *g* for 30 min at 4 °C. The supernatant containing PtmU4 was purified in three steps using an ÄKTA FPLC system (GE Healthcare Biosciences): (a) nickel-affinity chromatography equipped with a HisTrap HP, 5 mL column (GE Healthcare Life Sciences), which was first washed with 300 mL Wash buffer (50 mM Tris, pH 8.0, containing 300 mM NaCl and 20 mM imidazole) and then eluted with 100 mL 50% Elution buffer (50 mM Tris, pH 8.0, containing 100 mM NaCl and 500 mM imidazole); (b) anion exchange chromatography equipped with a HiTrap Q HP, 5 mL column (GE Healthcare Life Sciences) using a gradient increasing the concentration of sodium chloride from 0 to 1 M in 50 mM Tris, pH 8.0; (c) size-exclusion chromatography equipped with a Superdex 200 16/600 column (GE Healthcare Life Sciences) using a buffer of 50 mM Tris, pH 7.8, containing 100 mM NaCl, 50 mM KCl, and 5% glycerol. The resultant protein with an N-terminal His$_6$-tag was concentrated using an Amicon Ultra-15 concentrator (Millipore). Protein concentrations were determined from the absorbance at 280 nm using a molar absorptivity constant ($\varepsilon_{280} = 103,280\,M^{-1}$ cm$^{-1}$). Individual aliquots of PtmU4 were stored at –80 °C until use. Each of the PtmU4 site-directed mutants was produced and purified as described above.

**Enzymatic activity of PtmA3**. Preliminary incubations were performed in 50 mM phosphate, pH 7.6, containing 1 mM ATP, 1 mM CoA, 5 mM MgCl$_2$, 1 mM **5**, and 2 μM PtmA3 in a total volume of 50 μL. After incubation at 30 °C for 10 min, 50 μL of CH$_3$OH were added to quench the reaction. The reaction mixture was then centrifuged and 2 μL of the supernatant were injected and analyzed by LC-MS. Substrate and product were detected by monitoring 260 nm with a photodiode array detector. The reactions conditions for PtmA3 were optimized by monitoring **5-CoA** production using the HPLC method with a flow rate of 0.8 mL min$^{-1}$ and a 6 min solvent gradient from 2.5–20% CH$_3$CN in 10 mM ammonium acetate. Buffers (Tris and phosphate) and different concentrations of ATP, CoA, and MgCl$_2$ were all tested for improved PtmA3 activity. The optimized reaction conditions were determined to be 50 mM Tris, pH 8.0, containing 2.5 mM ATP, 2.5 mM CoA, and 5 mM MgCl$_2$, and used for the kinetic studies of PtmA3.

**Kinetic studies of PtmA3**. All kinetics assays were performed in the optimized reaction conditions with varying concentration of aryl acids (**5–10**) in a total volume of 50 μL. Each reaction was incubated at 30 °C for 10 min and boiled 1 min to quench the reaction. After centrifugation, the reaction mixtures were analyzed by HPLC as described above, but using different solvent gradients (**5** and **8**, 2.5–20% CH$_3$CN in 8 min; **6**, **7**, **9**, and **10**, 2.5–30% CH$_3$CN in 8 min) and the integrated area under curve (AUC) at 260 nm was calculated. A standard curve of **5-CoA–10-CoA** was used to convert AUC into the amount of product formed. Each kinetic assay was performed in triplicate.

**Enzymatic activity of PtmU4**. The reaction was first incubated in 50 mM phosphate, pH 7.4, containing 500 μM ATP, 2 mM MgCl$_2$, 2 mM Na$_2$S$_2$O$_3$, 100 μM PtmS2$^{GG}$ or 100 μM PtmS3$^{GG}$ and 40 μM PtmS4 at 30 °C for 30 min. Then, 100 μM **5-CoA** and 5 μM PtmU4 was added in a total volume of 50 μL. After incubation at 30 °C for another 30 min, the reaction was quenched by boiling for 1 min. The reaction mixture was then centrifuged and 20 μL of the supernatant were injected and analyzed by HPLC. Each sample was run on an Agilent 1260 HPLC system equipped with an Agilent Poroshell 120 EC-C18 column (50 mm×4.6 mm,

2.7 μm) using a 6 min solvent gradient (0.8 mL min$^{-1}$) of 2.5–20% CH$_3$CN in 10 mM ammonium acetate. Substrate and product were detected by monitoring 260 nm with a photodiode array detector.

When potassium hydrosulfide (KSH) was used to replace the native sulfur donor (sulfur-carrier protein), the reaction was performed in 50 mM phosphate, pH 7.4, containing 5 mM KSH, 200 μM **5-CoA**, and 5 μM PtmU4 in a total volume of 50 μL. After incubation at 30 °C for 10 min, the reaction was quenched by boiling for 1 min. The reaction mixture was then centrifuged and 10 μL of the supernatant were injected and analyzed by HPLC as described above. The relative activities of all PtmU4 mutants were determined using a **5-CoA** concentration of 500 μM. Due to slower turnovers of all mutants, enzyme concentration and incubation time were increased to 10 μM and 1 h, respectively, to facilitate product detection. The reaction mixture was then centrifuged and 10 μL of the supernatant were injected and analyzed by HPLC as described above. Substrate promiscuity assays of PtmU4 were determined using 500 μM of different CoA substrates and 0.5 μM PtmU4. The aryl thioacid peaks were collected and analyzed by LC-MS using either positive or negative mode (Supplementary Fig. 48).

The one-pot reaction to synthesize **5-SH** from **5** using a combination of PtmA3 and PtmU4 was performed using **5** (100 μM), CoA (50 μM), ATP (1 mM), Mg$^{2+}$ (4 mM), KSH (2 mM), PtmA3 (10 μM), and PtmU4 (5 μM) in phosphate buffer (50 mM, pH 7.4) at 30 °C.

**Antibiotic binding using surface plasmon resonance**. Experiments were performed on a Biacore X100 (GE Healthcare) instrument at 25 °C and data were analyzed using Biacore X100 evaluation software. HBS-P+buffer (0.1 M HEPES, 1.5 M NaCl, 0.5% v/v surfactant P20, pH 7.4) containing 0.1% dimethyl sulfoxide (DMSO) was used as the running buffer. Cells 1 and 2 were used as the reference and experimental surface, respectively. FabF C163Q was diluted to 0.15 μM in HBS-P+buffer containing 0.1% DMSO and immobilized on an NTA sensor chip (GE Healthcare) at a flow rate of 30 μL min$^{-1}$. For kinetic analysis, antibiotics (**1–4**) were prepared by two-fold serial dilutions with HBS-P+buffer containing 0.1% DMSO (0.078–10 μM) and injected over both surfaces at a flow rate of 30 μL min$^{-1}$. A 120-s association phase was followed by a 350-s dissociation phase. Signals from the reference surface and buffer blank injections were subtracted and the corrected results were globally fit to a 1:1 binding model. The association rate constant ($k_a$) and dissociation rate constant ($k_d$) were used to determine the equilibrium dissociation constant ($K_d$) in units of M.

**Computational details**. The crystal structure of the *E. coli* FabF C163Q–PTM complex[9] (PDB 2GFX) was used to extract the ADHBA moiety of PTM and the side chains of H303 and H340, or H303, H340, and Q163. For calculations with ADHBSH, one oxygen in the carboxylic acid group of ADHBA was replaced with a sulfur atom. Quantum mechanical DFT calculations were performed using Gaussian 09[45]. Each of the geometry optimizations were performed at the M06-2X/6-311+G(d,p) level of theory with the SMD implicit solvation model to account for the solvation effects of water and the interior of protein ($\varepsilon = 4$).

**Data availability**. Proteins from *S. platensis* CB00739 have been deposited to protein database of the National Center for Biotechnology Information (NCBI), under accession code AIW55578 for PtmA3, AIW55577 for PtmU4, AVR47602 for PtmS1, AVR47603 for PtmS2, AVR47604 for PtmS3, AVR47605 for PtmS4, AVR47606 for *Sp*SCP1, and AVR47607 for *Sp*SCP2. All other relevant data that support the findings of this study are available in the manuscript and the Supplementary Information.

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

## Acknowledgements

This work is supported in part by the Chinese Ministry of Education 111 Project B0803420 (to Y. Duan,) and National Institutes of Health Grants GM114353 (to B.S.). We thank Drs. Christoph Rader and Jinsai Shang, The Scripps Research Institute, for use of and expertise in using the SPR instrument, respectively, Dr. S.B. Singh, Merck Research Laboratories, Rahway, NJ, for providing *S. platensis* MA7327 and MA7339 wild-type strains, and the John Innes Center, Norwich, UK, for providing the REDIRECT technology kit. J.D.R. is supported in part by an Arnold O. Beckman Postdoctoral Fellowship. N.W. is supported in part by the Institute of Applied Ecology, Chinese Academy of Sciences, and a scholarship from the Chinese Scholarship Council (201504910034). This is manuscript #29626 from The Scripps Research Institute.

## Author contributions

B.S. conceived the project; L.-B.D., J.D.R., and B.S. designed the experiments; L.-B.D., J.D.R., D.K., N.W., and Y. Deng performed the experiments; L.-B.D., J.D.R., D.K., N.W., Y. Deng, Y.H., Y. Duan, and B.S. analyzed the results; C.Q.H. and K.N.H. performed the computational calculations; and L.-B.D., J.D.R., and B.S. wrote the manuscript with inputs from all co-authors.

## Additional information

**Competing interests:** The authors declare no competing interests.

