## [Peer Review File · Nature Communications]

Reviewers' comments:

Reviewer #1 (Remarks to the Author):

This is an excellent paper in which the authors identify the enzymes involved in thiocarboxylate formation in thioplatensimycin and thioplatencin and make a very important prediction: the thiocarboxylate functional group is likely to have been missed in many natural products because of its hydrolytic lability. The paper is suitable for publication in Nature Communications after consideration of the following points (all minor):

Page 29, line 516: None of the given MG accession numbers enabled me to retrieve the corresponding sequences. This should be corrected.

Page 32: Where is the ptmS2 gene located?

Page 35: Why is the elution order of compounds 1, 2, 3 and 4 different in panel C from the order shown in panels a and b?

Reviewer #2 (Remarks to the Author):

The manuscript describes the reconstitution of thioacid formation in thioplatensimycin and thioplatencin biosynthesis and provides the first clear evidence that the thiolated congeners are the final products of the biosynthetic cluster. While the biosynthesis of the thioacid moieties of other natural products has been postulated to occur using homologues of the proteins discussed in this paper (PtmA3 and PtmU4), the authors provide the first experimental evidence to validate this previous proposal.

The main novelty of the manuscript is the demonstration that a type III CoA transferase homologue is responsible for thioacid biosynthesis. Given this fact, a discussion of the activity of PtmU4 in the context of characterized type III CoA transferase homologues should be included. A supplemental figure detailing the activities attributed to characterized members would be useful, but is not required.

A reference to thiol trafficking literature (especially citation 21) should be included in the first portion of the paragraph beginning with line 71. I understand the reference to the genome sequence of the strain; however, it would be much more useful to a reader to provide a better explanation for the known sulfur trafficking pathways. Furthermore, the authors should mention

(or at least add a reference) to the work of Begley's group on thioquinolobactin in this paragraph. An explanation of the role that the JAMM protease plays in C-terminal cap removal should either be included in the main text (preferably) or in the SI with appropriate mention in the main text.

A few minor notes:

- Are homologues of PtmS1-S3 found in *S. avermitilis*? If so, this should be mentioned. If not, it is unclear why PtmS1 would be required for thioPTM/PTN biosynthesis as a sulfur carrier protein lacking the C-terminal extension could likely be used.
- Was 5-SH produced in the other strains that received the pBS12086 plasmid? Please include LCMS traces for these strains.
- Figure 1b would be improved if guidelines were included to help differentiate the thioacid peaks from the carboxylic acid peaks.
- Extended figure 1d: Although not necessary, the inclusion of the YTK, PDTC, and TQB clusters (with % Sim/ID values for the PtmA3 and PtmU4 homologues) would be a welcome addition.
- A short discussion of the difference between mono and di-domain CoA-transferases would be useful to help orientate readers outside the field.
- Please state the actual rate enhancement over background observed for the sulfur transfer of PtmU4 using the native sulfur donor.
- Extended data fig. 9: Please provide methods for how the stability assays were performed (especially since the elution order of the thioacid and carboxylic acid congeners is reversed).

The conclusions are supported by the experimental data and sufficient experimental details are provided. The findings will be of interest to the broader field of natural product biochemists and enzymologists. With the aforementioned minor changes I recommend publication of the manuscript.

Typos:

- Line 90: Based on the numbers in the table, the k_{cat}/K_m for 5 is 12.5-fold higher (not 25-fold) than for 6.
- Line 91: Sentence beginning with 5-CoA seems to have an error.
- Line 95: Should be "sulfide" not "disulfide"
- Line 435: "pH" missing
- Line 484: I think that "5-CoA" is a mistake.
- Line 609: Typo in "Thus"
- Extended data figure 2c: I think the peak labels are switched.

Reviewer #3 (Remarks to the Author):

The manuscript entitled "Biosynthesis of thiocarboxylic acid-containing natural products" by

Dong et al. describes the identification of two proteins that are involved in the biosynthesis of antibacterial natural products such as thioPTM and thioPTN. It is important to note that the identification of thioPTM/PTN was already described previously by the same research group in a prior publication.

This manuscript describes the identification of a "thio acid cassette" containing two genes that are responsible for the biosynthesis, and goes further in claiming that the thioacid congeners of PTM and PTN are the bona fide end products of the synthetic cascade. The authors attempt to draw extra significance to the identification of the "thioacid cassette" and the fact that it appears to be conserved across a wide spectrum of microbes - a finding that suggests that thioacid containing natural products are a heretofore underappreciated class of natural products.

On the positive side, the manuscript is well-written, well-referenced, and describes a large body of research.

This reviewer, however, does not feel that the paper rises to the level of a nature journal publication for the following reasons.

- 1) The most interesting aspect of this work could have been the identification of a new class of natural products (thioacids) with demonstrable biological activity. The manuscript does not deliver on this tantalizing claim - thioPTM and thioPTN had already been reported in the literature, and no evidence is presented that the seemingly conserved thioacid cassette is actually functionally integrated into the biosynthetic pathways of the organisms alluded to in the paper. Furthermore, no new thioacid containing molecules are described that would back up this claim. There are many examples of non-functional or deprecated genes in many organisms, so how do we know that thioacids have not already been evolutionarily selected against for some reason?
- 2) I find the claim that the thioacid congeners of PTM and PTN are the true end products of their synthesis is not proven satisfactorily, and in the worst case is basically irrelevant. Biosynthesis of natural products can be significantly influenced and altered under different growth conditions and passage/selection of strains. What is happening in *Streptomyces platensis* under "normal" wild-type growth conditions in a soil environment is not shown. Would that not be the true measure of the actual end product of biosynthesis? Is there some selective advantage under normal growth conditions in the wild for bacteria that could make thioPTM/PTN versus those that can't? Not shown, and this would have to be known to really assess the significance (or insignificance) of this observation.

The paper appears to be a collection of a bunch of data related to the PTM/PTN natural products that didn't fit into other papers already published by the group, loosely stitched together around a trumped up claim of significance related to thiocarboxylic acids. Worthy of publication

somewhere, but probably not in a Nature family journal. Had this paper focused on in depth characterization of the thioacid cassette genes and a survey of previously unknown thioacid containing natural products with interesting biological activity, I would be much more enthusiastic.

Responses to reviewer's comments

We are submitting the revised manuscript NCOMMS-18-03035-T, and detailed below are changes made during the revision. We thank you and the three reviewers for their excellent comments, criticisms, and suggestions regarding this submission and to you for your advice on how to prepare the revision. We have now addressed each of the reviewers' concerns by revising and updating the manuscript and Supplementary Information. The changes made in the manuscript, based on the referee's suggestions, are highlighted in the text, and the changes made in the Supplementary Information are not highlighted. We have also reformatted the manuscript according to *Nat. Commun.* style as the original manuscript was previously written as a letter. Summarized below are the specific changes made in revision.

Reviewer 1:

We thank reviewer 1 for their extremely supportive comments and address each of the requested minor concerns as detailed below:

1) *"Page 29, line 516: None of the given MG accession numbers enabled me to retrieve the corresponding sequences. This should be corrected."*

The accession numbers are now available. We also updated the accession numbers for each protein as the MG identifiers were for genes and not proteins (Page 22).

2) *"Where is the ptmS2 gene located?"*

A supplemental figure showing the genetic neighborhood of *ptmS2*, as well as *ptmS1*, *ptmS3*, and *ptmS4* has been included as Supplementary Fig. 8.

3) *"Why is the elution order of compounds 1, 2, 3 and 4 different in panel C from the order shown in panels a and b?"*

We apologize for this typo, which has now been corrected (Supplementary Fig. 2).

Reviewer 2:

We thank reviewer 2 for their extremely supportive comments and address each of the requested concerns as detailed below:

Reviewer 2's major concerns:

1) *"A discussion of the activity of PtmU4 in the context of characterized type III CoA transferase homologues should be included. A supplemental figure detailing the activities attributed to characterized members would be useful, but is not required."*

We have now included additions to the text, including a supplemental figure, discussing the activity of PtmU4 in the context of other characterized type III CoA transferases (Pages 8 and 9; Supplementary Fig. 26).

2) *"A reference to thiol trafficking literature (especially citation 21) should be included in the first portion of the paragraph beginning with line 71. I understand the reference to the genome sequence of the strain; however, it would be much more useful to a reader to provide a better explanation for the known sulfur trafficking pathways. Furthermore, the authors should mention (or at least add a reference) to the work of Begley's group on thioquinolobactin in this paragraph. An explanation of the role that the JAMM protease plays in C-terminal cap removal should either be included in the main text (preferably) or in the SI with appropriate mention in the*

main text.

We cited the thiol trafficking literature and added a brief introduction of the known sulfur trafficking pathways as suggested (Page 6). We also have now included additions to the text discussing the capped sulfur-carrier proteins and the role that the JAMM protease plays in removing these caps (Page 7). We also cited the work of Begley at this section.

Reviewer 2's minor concerns:

3) *"Are homologues of PtmS1-S3 found in S. avermitilis? If so, this should be mentioned. If not, it is unclear why PtmS1 would be required for thioPTM/PTN biosynthesis as a sulfur carrier protein lacking the C-terminal extension could likely be used."*

Yes, highly homologous proteins of PtmS1–S4 are found in *S. avermitilis*. This information including accession numbers and identity percentages is now added to the text and as Supplementary Table 4.

4) *"Was 5-SH produced in the other strains that received the pBS12086 plasmid? Please include LCMS traces for these strains."*

As stated in the text, **5-SH** was not produced in the other heterologous hosts. As requested, Fig. 1f now includes the LC-MS chromatograms for all heterologous hosts containing plasmid pBS12086.

5) *"Figure 1b would be improved if guidelines were included to help differentiate the thioacid peaks from the carboxylic acid peaks."*

Fig. 1 was updated as suggested. In addition, we expanded Fig. 1 from a one-column to two-column figure, to increase the size of the chromatograms.

6) *"Extended figure 1d: Although not necessary, the inclusion of the YTK, PDTC, and TQB clusters (with % Sim/ID values for the PtmA3 and PtmU4 homologues) would be a welcome addition."*

We included the protein sequence identities in the main text as suggested (Page 5).

7) *"A short discussion of the difference between mono and di-domain CoA-transferases would be useful to help orientate readers outside the field."*

The sequence-structure-function relationships of one vs. two domain CoA transferases is still unknown, but we have included a statement in the main text to clarify this for the non-experts (Pages 8 and 9).

8) *"Please state the actual rate enhancement over background observed for the sulfur transfer of PtmU4 using the native sulfur donor."*

This value (~3-fold) is now included in the main text (Page 9).

9) *"Extended data fig. 9: Please provide methods for how the stability assays were performed."*

As requested, the methods used to examine in vitro stability are now included in the legend of Supplementary Fig. 38.

Typos:

10) *"Line 90: Based on the numbers in the table, the k_{cat}/K_m for 5 is 12.5-fold higher (not 25-fold) than for 6."*

After recalculation, the number was changed to "12-fold higher than for **6**" (Page 8).

11) *"Line 91: Sentence beginning with 5-CoA seems to have an error."*

This sentence was revised (Page 8).

12) *“Line 95: Should be “sulfide” not “disulfide””*

Revised as suggested and for consistency, we used potassium hydrosulfide (KSH) instead of potassium hydrogen sulfide (KSH) throughout the paper (Page 8).

13) *“Line 435: “pH” missing”*

Revised as suggested (Page 19).

14) *“Line 484: I think that “5-CoA” is a mistake.”*

This sentence was revised (Page 21).

15) *“Line 609: Typo in “Thus””*

This typo was corrected (Supplementary Fig. 16).

16) *“Extended data figure 2c: I think the peak labels are switched.”*

This was a typo and has been corrected (Supplementary Fig. 2).

Reviewer 3:

We thank reviewer 3 for their constructive comments and we will address each of the requested concerns as detailed below:

1) *“The most interesting aspect of this work could have been the identification of a new class of natural products (thioacids) [. . .]. The manuscript does not deliver on this tantalizing claim – thioPTM and thioPTN had already been reported in the literature.”*

We agree that the discovery of a new thioacid-containing natural product with demonstrable biological activity would be an exciting discovery. Our work clearly depicts that two proteins encoded by a “thioacid cassette,” an acyl-CoA synthetase and an acyl-CoA transferase-like sulfur transferase, are responsible for the biosynthesis of the thioacid moiety of thioPTM and thioPTN. The characterization of this thioacid cassette, along with its prevalence in other bacteria, enables future discovery efforts of natural products with thioacid moieties.

As now described in the introduction for clarity, we previously identified a thioester-containing PTM pseudodimer, a presumed nonenzymatic result of a heteroatom Michael-addition of a proposed molecule of thioPTM with a molecule of PTM, as referred to by reviewer 3. The discovery of that pseudodimer gave the first indication that thioPTM may be produced within *Streptomyces platensis*. Further HRESI-MS and chemical transformations supported the presence of thiocarboxylic acid congeners of PTM and PTN, but thioPTM and thioPTN were not isolated in that study. The current work, which details not only the isolation and biological characterization of thioPTM and thioPTN, describes the biosynthesis of these two thioacid-containing natural products. Therefore, this project was intended to characterize this pathway and explore the potential implications of thioacid natural products.

2) *“No evidence is presented that the seemingly conserved thioacid cassette is actually functionally integrated into the biosynthetic pathways of the organisms alluded to in the paper. There are many examples of non-functional or deprecated genes in many organisms, so how do we know that thioacids have not already been evolutionarily selected against for some reason?”*

While we understand the reviewer’s concern that it is currently unknown how the additional thioacid cassettes are integrated into biosynthetic pathways, or even become non-functional due to evolution, the thioacid

cassette in thioPTM and thioPTN biosynthesis is undoubtedly functional and integrated in the *ptm* biosynthetic gene cluster. This is strongly supported by our (i) *in vivo* knockouts (where gene deletions of both the thioacid cassette and the sulfur-carrier protein machinery abolish production of thioPTM and thioPTN); (ii) fermentation of various *S. platensis* strains (all of which, including WT strains isolated from 3 different continents, produce thioPTM and/or thioPTN); and (iii) *in vitro* data (where PtmU4 is functional in the context of the native sulfur donors).

In addition to the evidence provided for the PTM/PTN pathways, the previously isolated thioacid- or thioester-containing natural products TQB, PDTC, and YTK and the presence of thioacid cassettes in their gene clusters, substantiates their functional integration in thioacid formation and biosynthesis of three additional natural products. If thioPTM and thioPTN was the only example, we would be tempted to agree that thioacid were evolutionarily selected against; however, four known examples from four different systems strongly support our conclusion that these cassettes are functionally integrated. Given our work, and the correlation between the TQB, PDTC, and YTK thioacid cassettes, we prefer to envision that the uncharacterized thioacid cassette, as with all unknown proteins, have important biological roles in nature that are awaiting discovery and characterization.

Finally, as described and referenced in the text, the antifungal activities and detoxification of the bacterial siderophores TQB and PDTC are higher than their carboxylic acid congeners, respectively, implicating a functional and biological role for thioacids in nature.

These ideas are summarized in the discussion section (Pages 11-13).

3) *“The claim that the thioacid congeners of PTM and PTN are the true end products of their synthesis is not proven satisfactorily, and in the worst case is basically irrelevant. Biosynthesis of natural products can be significantly influenced and altered under different growth conditions and passage/selection of strains. What is happening in Streptomyces platensis under ‘normal’ wild-type growth conditions in a soil environment is not shown. [. . .] This would have to be known to really assess the significance (or insignificance) of this observation.”*

Although we respectively disagree with the reviewer’s conclusion (see below), we have toned down the conclusion that thioPTM and thioPTN are the bona fide biosynthetic end products (Pages 2 and 4) and modified Fig. 2 to not include the label “biosynthetic end products.” Part of the discussion also addresses this idea (Pages 11 and 12).

We agree that laboratory conditions are seldom likely to exactly mirror those of the natural growth conditions in a soil environment; however, the detection of genuine natural growth conditions and to fully understand the selective advantages in these environments is technically impossible. With that being said, the genetic encoding of the thioacid cassette (*ptmA3* and *ptmU4*) within the biosynthetic gene cluster, our genetic knockouts of *ptmA3*, *ptmU4*, *ptmS1*, and *ptmS4* abolishing the production of thioPTM and thioPTN, and the clear production of thioPTM and thioPTN at day 1 (no PTM and PTN are produced at day 1), strongly support that the thioacid congeners are in fact the true, genetically encoded, final natural products. Any other explanation would be in direct conflict against the evident data. Finally, we thank reviewer’s 2 support that our work “provides the first clear evidence that the thiolated congeners are the final products of the biosynthetic gene cluster.”

4) *“The paper appears to be a collection of a bunch of data related to the PTM/PTN natural products that didn’t fit into other papers already published by the group, loosely stitched together around a trumped up claim of significance related to thiocarboxylic acids.”*

We regret that the reviewer came to this conclusion after reading it. In reality, after our discovery of the sulfur-containing pseudodimer, we explicitly designed these experiments to (i) support the legitimacy of the thioacid congeners as natural products, (ii) reveal the biosynthetic pathway of thioPTM and thioPTN, including the required sulfur trafficking pathway, (iii) determine whether the thioacid cassette was limited to the thioPTM and thioPTN system, (iv) highlight the potential of the thioacid cassette for biocatalysis, and (v) understand the biological implications of natural products containing thioacids.

Additional non-scientific changes:

As this manuscript was originally submitted to *Nature* and *Nature Chem. Biol.* and transferred to *Nat. Commun.* without style changes, the revised manuscript has been revised to comply with the formatting requirements for *Nat. Commun.* This resulted in rearrangements of the text, figures, tables, and supplementary information.

Reviewers' Comments:

Reviewer #2 (Remarks to the Author):

The authors made all of the changes I suggested. The paper is, in my opinion, is ready for publication.

Reviewer #3 put his/her comment in "Remarks to the Editor" section. (S)he thinks some of the previous concerns have been addressed. However, (s)he still concerns about the significance of the work in the absence of support for the claim that this pathway is responsible for a large number of heretofore undiscovered natural products containing thiocarboxylates. (S)he cannot believe that thiocarboxylates were missed for so long time if they truly represented an important class.

Responses to Reviewers:

It should be noted here that only the comment from reviewer 3 is addressed here, as reviewers 1 and 2 did not have any additional concerns. Responses to editorial and other requests are also summarized here.

Response to reviewer 3:

Reviewer 3 is concerned about “the significance of the work in the absence of support for the claim that this pathway is responsible for a large number of heretofore undiscovered natural products containing thiocarboxylates. [. . .] cannot believe that thiocarboxylates were missed for so long time if they truly represented an important class.”

We understand the reviewer’s concern given the nature of this surprisingly discovery (i.e., discovery that PTM and PTN are genetically encoded as thioPTM and thioPTN). Our characterization of the thioacid cassette, which is responsible for thiocarboxylic acid formation, was found in at least an additional 160 non-PTM- and -PTN-encoding genomes. While at this time we cannot definitively support that there are >160 thiocarboxylic acid-containing natural products awaiting discovery, the known natural products PDTC, TQB, and YTK support that this speculative conclusion is possible. Finally, as in all disciplines of science, the speed of discovery has no bearing on its importance.